# Influence of Sex and Strain on Hepatic and Adipose Tissue Trace Element Concentrations and Gene Expression in C57BL/6J and DBA/2J High Fat Diet Models

**DOI:** 10.3390/ijms232213778

**Published:** 2022-11-09

**Authors:** Kristen A. Hagarty-Waite, Melissa S. Totten, Matthew Pierce, Seth M. Armah, Keith M. Erikson

**Affiliations:** 1Department of Nutrition, University of North Carolina at Greensboro, Greensboro, NC 27412, USA; 2Department of Chemistry and Physics, Salem College, Winston-Salem, NC 27101, USA

**Keywords:** high fat diet, adipose, gene expression, trace elements, macrophage

## Abstract

The objective of this study was to determine the influence of sex and strain on the dysregulation of trace element concentration and associative gene expression due to diet induced obesity in adipose tissue and the liver. Male and female C57BL/6J (B6J) and DBA/2J (D2J) were randomly assigned to a normal-fat diet (NFD) containing 10% kcal fat/g or a mineral-matched high-fat diet (HFD) containing 60% kcal fat/g for 16 weeks. Liver and adipose tissue were assessed for copper, iron, manganese, and zinc concentrations and related changes in gene expression. Notable findings include three-way interactions of diet, sex, and strain amongst adipose tissue iron concentrations (*p* = 0.005), adipose hepcidin expression (*p* = 0.007), and hepatic iron regulatory protein (IRP) expression (*p* = 0.012). Cd11c to Cd163 ratio was increased in adipose tissue due to HFD amongst all biological groups except B6J females, for which tissue iron concentrations were reduced due to HFD (*p* = 0.002). Liver divalent metal transporter 1 (DMT-1) expression was increased due to HFD amongst B6J males (*p* < 0.005) and females (*p* < 0.004), which coincides with the reduction in hepatic iron concentrations found in these biological groups (*p* < 0.001). Sex, strain, and diet affected trace element concentration, the expression of genes that regulate trace element homeostasis, and the expression of macrophages that contribute to tissue iron-handling in adipose tissue. These findings suggest that sex and strain may be key factors that influence the adaptive capacity of iron mismanagement in adipose tissue and its subsequent consequences, such as insulin resistance.

## 1. Introduction

From 1999 to 2018, the prevalence of adult obesity and severe obesity has increased in the United States, reaching an age-adjusted prevalence of 42.1% in 2017–2018 [1]. Obesity is associated with all-cause mortality and comorbidities such as neurodegenerative disease, non-alcoholic fatty liver disease (NAFLD), and type II diabetes mellitus [2,3]. Diet induced obesity (DIO) has also been associated with alterations to trace element homeostasis in the brain, liver, and adipose tissue, which has been demonstrated to contribute to the development of these comorbidities [4,5,6]. For example, imbalance of copper and zinc are associated with the development of Alzheimer’s Disease and the mismanagement of trace elements, particularly iron, in adipose tissue has been associated with insulin resistance potentially via an adiponectin-dependent pathway [7,8,9]. Further, in the context of DIO-associated NAFLD, metabolic stress provoked by the accumulation of fatty acids in the liver can lead to alterations in hepatic and systemic trace element concentration and hepatic insulin resistance [4,6,10]. Given the commonality of several trace element transport mechanisms, like divalent metal transporter 1 (DMT-1) and ceruloplasmin, the alteration in homeostasis of any one trace element has the potential to dysregulate copper, iron, manganese, or zinc, leading to pathological consequences [11,12]. The effects of DIO or high fat diet (HFD) on trace element homeostasis in the liver and adipose tissue have been primarily in male C57 BL/6J mice. Our objective of this study is to extend these previous findings by exploring the effects of sex and strain on the influence of a HFD on systemic trace element biology.

The biological factors of genetics and sex are important considerations when exploring DIO and its associated comorbidities. The inclusion of male and female subjects is a key feature of an NIH initiative aimed to improve reliability in human and vertebrate animal research [13]. Additionally, there have been demonstrated differences in gene expression amongst strains of mice as a result of a HFD. A study investigating the impact of HFD on the endogenous antioxidant, glutathione, demonstrated differences amongst DBA/2J (D2J) and C57BL/6J (B6J) strains [14]. The B6J and D2J strains are both validated models of DIO [15,16,17]. In order to address the factors of sex and strain, our model of DIO includes male and female B6J and D2J mice.

Our lab has previously demonstrated that 16 weeks of a HFD led to the dysregulation of trace element concentration and gene expression in several regions of the brain [5,18,19]. Furthermore, we have established that there are sex and strain differences in trace element concentration and gene expression as a result of HFD within these brain regions amongst male and female B6J and D2J mice. For example, copper concentration in the hippocampus has been found to be influenced by HFD and strain, with increased and decreased copper concentrations in the B6J and D2J strains, respectively. We have also found that diet, sex, and strain impacted gene expression in this brain region as a HFD led to increased expression of DMT-1 and reduced expression of ceruloplasmin for B6J males [5]. However, the influence of sex and strain on the systemic distribution of trace elements and gene expression in a model of DIO, remains unexplored and valuable in the context of metabolic conditions associated with altered trace element homeostasis, such as tissue insulin resistance.

The main goal of this study was to determine the influence of sex and strain on the dysregulation of trace element concentration and gene expression due to HFD in adipose tissue and the liver. Recombinant inbred (RI) strains are an important resource for phenotypic and genotypic analysis of complex traits in many species. The BXD strain set, generated by crossing C57BL/6J (B6J) and DBA/2J (D2J) inbred strains, is one of the largest mouse reference panels [20]. The cumulative and integrative power of the BXD strain set makes it a valuable tool for the investigation of complex multigenetic relationships and analysis of genetic variability under different environmental conditions at various levels [21,22]. Previous studies have demonstrated that adipose and hepatic tissue are both highly susceptible to chronic inflammation in the setting of a HFD [23,24,25]. Further, HFD-associated inflammation in adipose tissue and the liver is associated with the development of insulin resistance [26,27]. Our primary hypothesis was that a HFD would lead to alterations in copper, iron, manganese, and zinc concentration and associative gene expression in the liver and adipose with main effects of sex and strain compared to normal-fat diet (NFD) counterparts.

## 2. Results

### 2.1. Weight Gain and Feed Efficiency

At the initiation of the 16-week dietary intervention, there was no significant difference in weight between NFD and HFD groups. However, following 16 weeks of high fat dietary treatment, both B6J and D2J males and females gained a significant amount of weight compared to normal fat counterparts (Figure 1).

There was no significant difference in total daily grams of food eaten between NFD (2.536 ± 0.072 g/day) and HFD (2.455 ± 0.049 g/day) groups (t(20) = 0.940, *p* = 0.358). Differences in tissue trace element concentration and gene expression were not due to differences in intake nor feed efficiency rate as there were no significant differences between the average feed efficiency rate amongst NFD (6.559 ± 5.316 kcal/g weight gained) and HFD (8.021 ± 1.367 kcal/g weight gained) groups, B6J (10.021 ± 4.461 kcal/g weight gained) and D2J strains (4.560 ± 2.996 kcal/g weight gained), and between NFD and HFD groups of each biological group (Table 1).

### 2.2. Effect of Diet, Sex, and Strain on Trace Element Concentration

#### 2.2.1. Liver

The homeostasis of iron, copper, and manganese were significantly impacted by HFD in the liver. Hepatic iron concentrations were reduced amongst all sex and strain combinations following a HFD (Figure 2). The reduction of iron concentration due to diet was statistically significant for both B6J males (F(1,35) = 19.329, *p* < 0.001) and females (F(1,35) = 68.677, *p* < 0.001). Hepatic iron was reduced by 51% for B6J males and 57% for B6J females. However, there were no significant diet-induced changes in iron concentration for D2J mice (*p* = 0.890). There were significant two-way interactions between strain and sex (F(1,35) = 7.510, *p* = 0.01) as well as sex and diet (F(1,35) = 6.394, *p* = 0.016) for liver copper concentrations. There was a simple main effect of diet for D2J males (F(1,35) = 15.747, *p* = 0.001). Hepatic copper concentrations were decreased by 24% due to HFD in this group. Hepatic zinc concentration was not significantly impacted by HFD. However, there was a significant interaction between sex and strain for hepatic zinc concentration (F(1,36) = 29.688, *p* < 0.001) with significant simple main effect of strain for both males (F(1,33) = 22.582, *p* < 0.001) and females (F(1,33) = 8.964, *p* = 0.005) and significant simple main effect of sex for B6J mice (F(1,33) = 47.087, *p* < 0.001). Interestingly, D2J males and females demonstrated a HFD-induced reduction in hepatic zinc concentrations, however, this was not statistically significant. There was a significant three-way interaction between diet, sex, and strain (F(1,31) = 12.310, *p* = 0.001) for hepatic manganese concentration with simple main effect of diet for D2J females (F(1,31) = 45.486, *p* < 0.001), demonstrating a significant 36% increase in manganese concentration due to a HFD in this group.

#### 2.2.2. Adipose Tissue

There was a statistically significant three-way interaction between diet, sex, and strain for adipose tissue iron concentrations (F(1,28) = 9.302, *p* = 0.005) with a significant two-way interaction between diet and sex for the B6J strain (F(1,28) = 9.032, *p* = 0.006). There was a significant simple main effect of diet for B6J females, for which adipose iron concentrations were reduced by 40% (F(1,28) = 11.462, *p* = 0.002). There was a significant main effect of strain for adipose copper concentration, with the B6J strain expressing copper at 160% greater than the concentration of D2J strains. While adipose zinc concentrations followed a similar pattern to liver zinc concentration for most groups, there were no significant interactions of diet, sex, and strain for adipose zinc concentrations. Similarly, there were no significant interactions of diet, sex, and strain for adipose manganese concentration. However, HFD did influence concentrations for B6J (t(7) = 2.536, *p* = 0.039) and D2J females (t(7) = 3.170, *p* = 0.016), demonstrating a reduction in adipose manganese concentration as a result of a HFD.

### 2.3. Effect of Diet, Sex, and Strain on the Gene Expression of Trace Element Homeostatic Proteins

#### 2.3.1. Liver

The expression of mRNA in adipose and liver tissue was evaluated for the following trace element homeostatic proteins: DMT-1, CTR-1, ceruloplasmin, IRP-1, hepcidin, and HIF1-α. In liver tissue, strain significantly impacted the expression of all genes evaluated (Figure 3). Additionally, the impact of a HFD on the expression of genes involved in iron homeostasis, like hepcidin and IRP, was significantly different amongst males and females. Specifically, there was a significant two-way interaction between sex and strain for hepcidin expression (F(1,31) = 10.143, *p* = 0.003), which was significantly decreased due to HFD in B6J males (F(1,31) = 30.464, *p* < 0.001). There was a significant three-way interaction between diet, sex, and strain for the expression of IRP (F(1,30) = 7.097, *p* = 0.012) with a significant simple main effect of diet for B6J females (F(1,30) = 16.375, *p* < 0.001). The expression of DMT-1 was significantly decreased due to HFD for both B6J males (t(8) = 3.793, *p* = 0.005) and females (t(8) = 3.997, *p* = 0.004). The expression of Ferroportin was not impacted by HFD, however, there were significant differences between male and female expression of ferroportin amongst both B6J (F(1,35) = 4.761, *p* = 0.036) and D2J (F(1,35) = 44.924, *p* < 0.001) strains. HFD also impacted copper-related gene expression in the liver. CTR-1 expression was significantly reduced due to HFD, by 2-fold for B6J females (*p* = 0.004) and the expression of ceruloplasmin was significantly decreased for both D2J females (*p* = 0.035) and B6J males (*p* = 0.007) due to a HFD.

#### 2.3.2. Adipose Tissue

Diet, sex, and strain also significantly impacted trace element-related gene expression in adipose tissue (Figure 4). There was a significant two-way interaction between sex and strain (F(1,30.015) = 50.889, *p* <0.001) for the expression of HIF1- α. HIF1- α expression was significantly higher amongst B6J female fed a HFD compared to B6J males fed a HFD (*p* < 0.001). D2J males had significantly higher HIF1- α expression compared to D2J females comparing both NFD (*p* = 0.007) and HFD groups (*p* < 0.001). There was a significant three-way interaction between diet, sex, and strain (F(1,30) = 10.64, *p* = 0.007) for the expression of hepcidin with simple main effect of strain for males that received a HFD (F(1,30) = 13.715, *p* < 0.001). D2J males that received a HFD demonstrated a 3.6-fold increase in expression of hepcidin compared to NFD controls, whereas B6J males demonstrated a 0.06-fold decrease in hepcidin expression due to a HFD. There was also a simple main effect of sex for the expression of hepcidin amongst B6J mice fed a HFD (F(1,30) = 14.391, *p* < 0.001). B6J females exhibited a 59-fold increase in hepcidin expression as a result of a HFD. However, the biological variability amongst this group likely contributed to the insignificant effect of diet amongst B6J females. There was a significant two-way interaction between diet and strain (F,(1,32) = 5.062, *p* = 0.031) for the expression of ferroportin. There was a significant simple main effect of strain for both male (*p* = 0.0015) and female (*p* = 0.0074) HFD groups with the B6J strain expressing ferroportin significantly greater than the D2J strain as a result of a HFD. There were no significant differences between NFD and HFD groups for the expression of IRP-1, DMT-1, and CTR-1. However, sex and strain significantly impacted the expression of DMT-1 and CTR-1, respectively. The expression of DMT-1 was significantly higher amongst males (95% CI −3.257 to −0.424, *p* = 0.012) and the expression of CTR-1 was significantly higher amongst the B6J strain (95% CI −3.346 to −0.237, *p* = 0.013). Sex and strain further impacted the expression of copper-related genes. There was a significant two-way interaction between sex and strain (F(1,31) = 7.142, *p* = 0.012) for the expression of ceruloplasmin with simple main effect of strain for females (F(1,31) = 11.827, *p*= 0.002) and simple main effect of sex for D2J mice (F(1,31) = 23.438, *p* < 0.001). Expression was statistically lower for D2J females compared to D2J males (95% CI, 2.212 to 5.432, *p* < 0.001) and significantly higher for B6J females compared to D2J females (95% CI, 1.105 to 4.325, *p* = 0.002).

### 2.4. Effect of Diet, Sex, and Strain on the Gene Expression of Repair (M2-like) and Pro-inflammatory (M1-like) Macrophages

#### 2.4.1. Liver

Macrophages have been demonstrated to play an integral role in local tissue iron management [28]. In order to investigate the impact of macrophages on HFD-induced changes in iron concentration in the liver and adipose tissue, we evaluated the expression of Cd11c, a marker of M1-like macrophages, Cd163, a marker of M2-like macrophages, and the ratio of M1 to M2-like macrophage expression. There were no significant differences in expression of Cd11c and Cd163 in the liver as a result of a HFD (Figure 5). However, sex and strain were significant factors influencing the expression of both genes. We found a significant two-way interaction between strain and sex (F(1,31) = 235.948, *p* < 0.001) for the expression of Cd11c with simple main effect of strain for both males (F(1,31) = 272.236, *p* < 0.001) and females (F(1,31) = 25.529, *p* < 0.001) and simple main effect of sex for D2J mice (F(1,31)= 418.308, *p* < 0.001). Additionally, B6J mice (−0.623–0.417) had significantly higher expression of Cd163 than D2J mice (t(37)= 2.405, *p* = 0.024) while male mice had significantly lower expression of Cd163 compared to female mice (t(25.4) = −3.738, *p* < 0.001). Expression of Cd11c to Cd163 ratio was significantly increased in B6J males (t(7) = −2.586, *p* = 0.036) and significantly decreased in D2J females (t(8) = 3.048, *p* = 0.016) due to HFD.

#### 2.4.2. Adipose Tissue

Using the established markers for both M1 and M2-like macrophages, we found that the expression of Cd11c was significantly increased as a result of a HFD for both B6J male mice (mean difference 6.557, 95% CI 1.165 to 9.243, t(8) = 5.629, *p* < 0.001) and D2J female mice (mean difference 1.951, 95% CI 0.720 to 3.183, t(8) = 3.654, *p* = 0.006) (Figure 5). The expression of Cd11c was also increased as a result of a HFD for B6J females and D2J males, but this was not statistically significant. There were no significant changes in expression of Cd163 amongst any group, as a result of a HFD. However, the expression of Cd163 was significantly higher amongst the D2J mice when compared to mice of the B6J strain (t(38) = 13.292, *p* < 0.001). HFD significantly influenced the ratio of M1-like to M2-like macrophages in adipose tissue for all groups except B6J females, consistent with the impact of HFD on adipose tissue iron concentrations. The ratio of Cd11c to Cd163 expression was significantly increased due to HFD for B6J males (95% CI −74.150 to −16.411, t(4.005) = −4.352, *p* = 0.012), D2J males (95% CI −10.098 to −5.240, t(8)= −7.281, *p* < 0.001), and D2J females (95% CI −18.958 to −2.241, t(8) = −2.924, *p* = 0.019), but not B6J females.

## 3. Discussion

The goal of this study was to determine the influence of sex and strain on the dysregulation of trace element concentration and gene expression due to a HFD in adipose tissue and the liver. Following 16 weeks of a HFD, we found that sex and strain were often key factors that influenced trace element concentration and gene expression in adipose and liver tissue, which is consistent with our previous findings in the brain [5].

The B6J and D2J strains, both established models of DIO, are the parent strains of the BXD recombinant inbred strains [29]. We have demonstrated differences in trace element concentration, regulation, and resultant macrophage recruitment in adipose and the liver of these two strains, in response to a HFD. These findings support the need for the use of multiple BXD strains in future studies in order to elucidate the individual genetic-based diversity in metabolic disease severity and susceptibility as a result of the HFD-induced dysregulation of trace elements in these tissues.

Our novel findings include the influence of HFD, sex, and strain on iron concentration, hepcidin expression, and Cd11c expression in adipose tissue as well as manganese concentration and IRP expression in the liver. Further, we found that sex, strain, and diet affected Cd11c:Cd163 expression in adipose and liver tissue, with the ratio for B6J females not influenced by HFD in either liver or adipose tissue. We providing a working model highlighting our novel findings (Figure 6).

### 3.1. HFD Impacted Trace Element Concentrations Differentially Due to Sex and Strain

Trace metal concentrations in both adipose and liver tissue were impacted by sex and strain due to HFD. Adipose manganese concentrations were significantly reduced following a HFD for both B6J and D2J females, but increased for D2J males. Hepatic zinc concentrations were significantly increased for both males and females of the B6J strain and significantly decreased for males and females of the D2J strain due to a HFD. We found that iron concentrations in the liver were reduced by 49% and 43% for B6J males and females, respectively. A study by Chung et al., utilizing a larger sample size of 16–18 mice per group, demonstrated a 30% reduction in hepatic iron concentration as well as a significant reduction in hepatic hepcidin expression in B6J male mice following 16 weeks of a 60% kcal from fat HFD starting at 4 weeks of age [30]. We have also demonstrated a significant reduction in hepatic hepcidin expression in B6J males, but not B6J females following a HFD. In rodent models, sex and strain are known to influence hepcidin expression in the liver and it has been demonstrated that in inbred strains, hepatic iron concentrations are not predictive of hepcidin expression [31]. Our study demonstrates similar findings in the setting of a HFD, supporting a more complex and varied response to iron regulation in the liver, further supporting the need for investigation into BXD inbred strain differences in this context. Hepcidin is a negative regulator of iron absorption by means of ferroportin, the only known iron exporter, degradation [32,33,34]. HFD-associated inflammation also impacts hepcidin expression and iron homeostasis, ultimately leading to iron dysregulation, which has been associated with metabolic disease, type II diabetes mellitus, and cardiovascular disease [35,36]. Interestingly, adipose iron concentrations were increased due to a HFD for all biological groups except for B6J females, for which iron concentrations were significantly reduced as a result of a HFD. Previous studies have demonstrated increased hepcidin expression in visceral adipose tissue following 24 weeks, but not 12 weeks, of a 60% kcal from fat HFD in Swiss strain mice [37]. We found, that despite the reduction in adipose iron concentration in B6J females, this group demonstrated a 59-fold increase in hepcidin expression in adipose tissue, further demonstrating the sex and strain differences in response to a HFD. Hepcidin mRNA expression has been found to be significantly increased in obese patients and is correlated with adiposity, independent of obesity-induced diabetes mellitus or non-alcoholic fatty liver disease [38].

### 3.2. Sex and Strain Impacted Relative Expression of Iron-Related Genes in Liver and Adipose Tissue

In adipose and liver tissue, the expression of genes that regulate trace element homeostasis was also influenced by sex and strain in the setting of a HFD. Previous studies have demonstrated increased expression of DMT-1 in response to HFD-induced systemic iron deficiency in B6J males [39,40,41]. In a study investigating the duodenal gene expression differences amongst several strains, including B6J and D2J mice, there were significant strain differences in hepatic iron as well as duodenal DMT-1 and ferroportin expression [41]. We found that hepatic DMT-1, IRP, and CTR-1 expression was significantly increased for female B6J mice following 16 weeks of a HFD. DMT-1 expression was also significantly increased in the liver for B6J males due to a HFD while hepatic IRP expression was also significantly increased due to a HFD in D2J males, likely in part related to the significant decrease in liver iron concentrations amongst the B6J strain. In adipose tissue, we found large biological variability between groups. Adipose DMT-1, IRP, and ferroportin expression were influenced by sex and/or strain, but not diet. Due to this, we investigated the role of macrophage polarization in order to better explain our findings on adipose iron concentration.

### 3.3. Macrophage Polarization May Explain Sex and Strain Differences in Adipose Iron Concentrations

A critical function of adipose tissue macrophages (ATMs) is the regulation of iron status in the local tissue environment [42]. However, HFD-induced inflammatory cytokine recruitment has been demonstrated to drive macrophage recruitment and polarization from a M2, repair-like phenotype, to a M1, inflammatory phenotype, which subsequently alters the iron handling capability of ATMs [43,44,45]. While ATMs are highly phenotypically plastic, the shift from predominately M2-like ATMs to M1-like in the setting of a HFD reduces the capacity for iron handling, leading to iron overload in the adipocyte, which has been associated with local and systemic insulin resistance [28,46,47,48]. Cd163, a transmembrane glycoprotein expressed exclusively in macrophages, mediates the uptake of haptoglobin-hemoglobin complexes and contributes to the management of local tissue iron concentrations, consistent with M2-like ATM function [49,50]. Expression of M1 surface protein, Cd11c, contributes to HFD-related inflammatory pathway activation and insulin resistance [51]. Therefore, Cd11c has been used as a marker to define M1-like ATMs [51,52,53,54]. Congruent with previous findings, we have demonstrated that HFD leads to increased expression of Cd163 and Cd11c, suggesting increased M2 and M1-like ATMs in B6J male mice [45]. However, there are significant strain and sex differences in the expression of M2 and M1-like ATMs. Comparable to B6J males, the expression of M1-like ATMs was significantly increased due to HFD in D2J females, although the expression of both M2 and M1-like ATMs was relatively low compared to mice of the B6J strain. B6J females demonstrated an increase in M1and M2-like ATMs as a result of HFD, but this was not significant. Interestingly, a HFD resulted in decreased expression of M1 and M2-like ATMs for D2J males. Further the ratio of Cd11c to Cd163 expression was influenced by diet, sex, and strain. Previous findings have demonstrated that HFD increases the expression of macrophages with low iron handling capacity and increases the polarization of macrophages with high iron handling capability to an M1-like phenotype, leading to impaired iron homeostasis and iron accumulation in adipose tissue of male B6J mice [45]. We found that HFD increased the ratio of M1 to M2-like ATMs in all groups except B6J females, who had significantly lower tissue iron concentrations. These changes in expression and ratio of differentially polarized macrophages in the context of a HFD may help to explain the changes in adipose iron concentration due to a HFD. Further, these findings suggest that sex and strain may be key factors that influence the ability to respond to HFD-induced iron mismanagement in adipose tissue and its resultant consequences.

## 4. Methods

### 4.1. Animals and Diet

Animal care and dietary intervention was conducted as previously described [5,18]. Briefly, male and female mice from the C57BL/6J (B6J) and DBA/2J (D2J) strains (n = 72) were purchased from the Jackson Laboratory (Bar Harbor, ME, USA) at post-natal day 21. Following a three-day acclimation period, mice were randomly assigned to a control, NFD containing 10% kcal fat/g (D12450B; Research Diets), or a mineral-matched HFD containing 60% kcal fat/g (D12492; Research Diets) for 16 weeks as previously described [5]. Treatment groups were as follows: B6J male NFD, B6J male HFD, B6J female NFD, B6J female HFD, D2J male NFD, D2J male HFD, D2J female NFD, and D2J female HFD (n = 9 per group). Mineral formulation (S10026; Research Diets) used for all groups met the minimum micronutrient requirements for rodents [55]. For the duration of the study, ad libitum feeding of the randomly assigned diet and free access to deionized water 24-h/day was provided. Ear notching was used to identify mouse number and mice were housed in groups of three per cage with male and female cages positioned on opposite sides of a 25 ± 1 °C temperature-controlled room. Mice were maintained on a 12 h light/dark cycle. Mice were weighed at the time of prescribed diet initiation and weekly until tissue collection. All animal care and procedures were performed with approval obtained from the Institutional Animal Care and Use Committee of the University of North Carolina at Greensboro and conformed to the guidelines established by the National Institutes of Health for the ethical care and use of laboratory animals.

### 4.2. Tissue Collection

At the conclusion of the 16-week dietary intervention, mice were euthanized by rapid decapitation following anesthetization with isoflurane. Blood, isolated plasma, liver, and visceral adipose tissue were collected and flash frozen in liquid nitrogen, placed on dry ice, and then stored at −80 °C until further processing.

### 4.3. Trace Element Concentration Analysis

Trace element concentration of adipose and liver samples was determined using graphite furnace atomic absorption spectrometry (Varian AA240, Varian, Inc., Palo Alto, CA, USA). Copper, iron, manganese, and zinc concentrations are reported as micrograms of trace element per gram of protein. Total protein concentration was determined using a Pierce Bicinchoninic Acid (BCA) Protein Assay (Thermo Fisher Scientific Inc., Waltham, MA, USA). A sample size of n = 9 per biological group was used for tissue trace element concentration analysis. Tissues samples were sonicated in cold radio-immunoprecipitation assay buffer (RIPA) containing protease inhibitors. Homogenates were digested in ultrapure nitric acid for 24 h in a sand bath maintained at 60–80 °C and used for protein analysis. Separately, 50 μL aliquots of the digested homogenate were further diluted in 2% nitric acid and utilized for trace element concentration analysis. Bovine liver containing 184 μg iron/g was digested in ultrapure nitric acid and used as an internal standard for trace element concentration analysis. All samples and controls were analyzed in duplicate apart from iron concentration, which was analyzed in triplicate.

### 4.4. RNA Isolation and cDNA Synthesis

Following manufacturer’s protocol, RNA was isolated from liver and adipose tissue samples (n = 5 per group) utilizing the RNeasy Plus Mini Kit and RNeasy Lipid Tissue Mini Kit (Qiagen Inc., Germantown, MD, USA), respectively. Concentration and purity of RNA was determined with a Nanodrop 1000 spectrophotometer (Thermo Fisher Scientific, Inc., Waltham, MA, USA). Reverse transcription for cDNA synthesis was conducted on Applied Biosystems GeneAmp PCR System 9700 using Applied Biosystems High-Capacity cDNA Reverse Transcription Kit (Life Technologies, Carlsbad, CA, USA) in a 20-µL reaction volume. cDNA samples were immediately used or stored at −20 °C.

### 4.5. Real Time Polymerase Chain Reaction (RT-PCR)

Relative gene expression was determined by RT-PCR using a 7500 Fast Real-Time PCR System from Applied Biosystems under thermal conditions as follows: 120-s incubation at 50 °C, 120-s polymerase activation at 95 °C, and 40 cycles of PCR consisting of 3-s denature at 95 °C and 30-s anneal/extend at 95 °C. TaqMan gene expression assay probes from Life Technologies (Carlsbad, CA, USA) were used to assess relative gene expression for DMT-1, ceruloplasmin, high affinity copper uptake protein 1 (CTR-1), hypoxia inducible factor 1 (Hif1-α), iron regulatory protein (IRP-1), hepcidin, ferroportin, Cd163 (M2-like macrophage), and Cd11c (M1-like macrophage). PCR reactions were performed in duplicate and the expression of each gene was normalized to the housekeeping gene 18S as the endogenous control. Fold change was determined using the 2^−ΔΔCT^ method and expressed as normalized to the B6J male NFD group unless otherwise denoted.

### 4.6. Statistical Analysis

Gene expression was assessed using normalized cycle threshold (Ct) values. Normality was confirmed using the Shapiro–Wilk test and homogeneity of variance was determined using Levene’s test. For normally distributed variables, means ± SEM were reported, and three factor analysis of variance (ANOVA) was used to determine the effect of diet, sex, and strain on trace element concentration and gene expression. Simple main effects were further investigated for statistically significant interactions. Statistically significant main effects were reported for data in which no statistically significant interactions were determined. If homogeneity of variances could not be achieved to perform three factor ANOVA, differences between groups was assessed using two-way independent *t*-tests. For variables that were not normally distributed, the medians (25th, 75th percentiles) were reported and aligned rank transformation (ART), a non-parametric technique, was performed prior to three factor ANOVA [56]. Statistical significance was set at *p* < 0.05. For pairwise comparisons using transformed non-parametric data, a Bonferroni correction for seven comparisons was applied and therefore, statistical significance was accepted at *p* < 0.0125. Statistical analysis was conducted using IBM SPSS Statistics 26 with the exception of ART ANOVA, which was performed using R software version 4.2.0 [57].

## 5. Conclusions

Our data show that diet, sex, and strain have a significant influence on the trace element concentrations and associated gene expression in adipose and liver tissue. A major finding is that iron concentrations and genes involved in iron regulation were particularly impacted (Summarized in Figure 6). In adipose tissue, the marked influence of sex, strain, and diet on M1-like and M2-like macrophage expression, may be able to explain differences in adipose tissue iron concentrations due to a HFD. Potential hormone-gene interactions may be the key to understanding the adaptive capacity of tissue iron management in the setting of a HFD, providing insight into the subsequent metabolic consequences of the dysregulation of iron homeostasis, including local tissue as well as systemic insulin resistance. Future studies will use this data to continue to explore the interaction between hormones and genetics as well as explain sex and strain-specific mechanisms of altered trace element homeostasis in the context of a HFD. Further elucidation of these mechanisms may also help to identify gene regulatory networks responsible for trace element dysregulation in the setting of a HFD and their role in the development of related chronic disease.

## Figures and Tables

**Figure 1 ijms-23-13778-f001:**
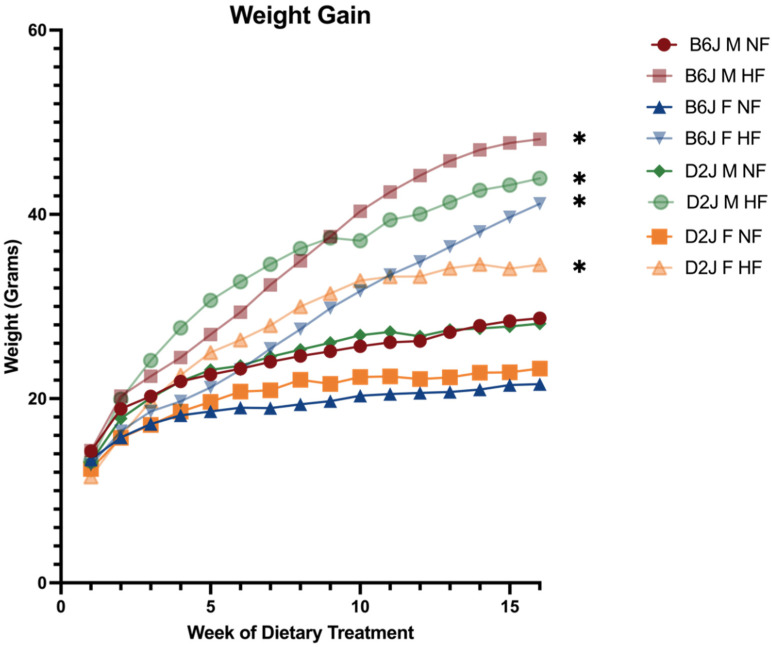
Average body weight over 16-week diet intervention. Comparisons between normal fat and high fat diet groups were conducted for B6J and D2J males and females at the beginning (week 1) and end of diet intervention (week 16). There were no significant differences in initial body weight between normal fat and high fat groups. Following a 16-week diet intervention, there were significant differences in weight between normal fat and high fat groups for males and females of both B6J and D2J strains. * *p* < 0.05. B6J M LF = B6J male normal fat, B6J M HF = B6J male high fat, B6J F LF = B6J female normal fat, B6J F HF = B6J female high fat, D2J M LF = D2J male normal fat, D2J M HF = D2J male high fat, D2J F LF = D2J female normal fat, D2J F HF = D2J female high fat.

**Figure 2 ijms-23-13778-f002:**
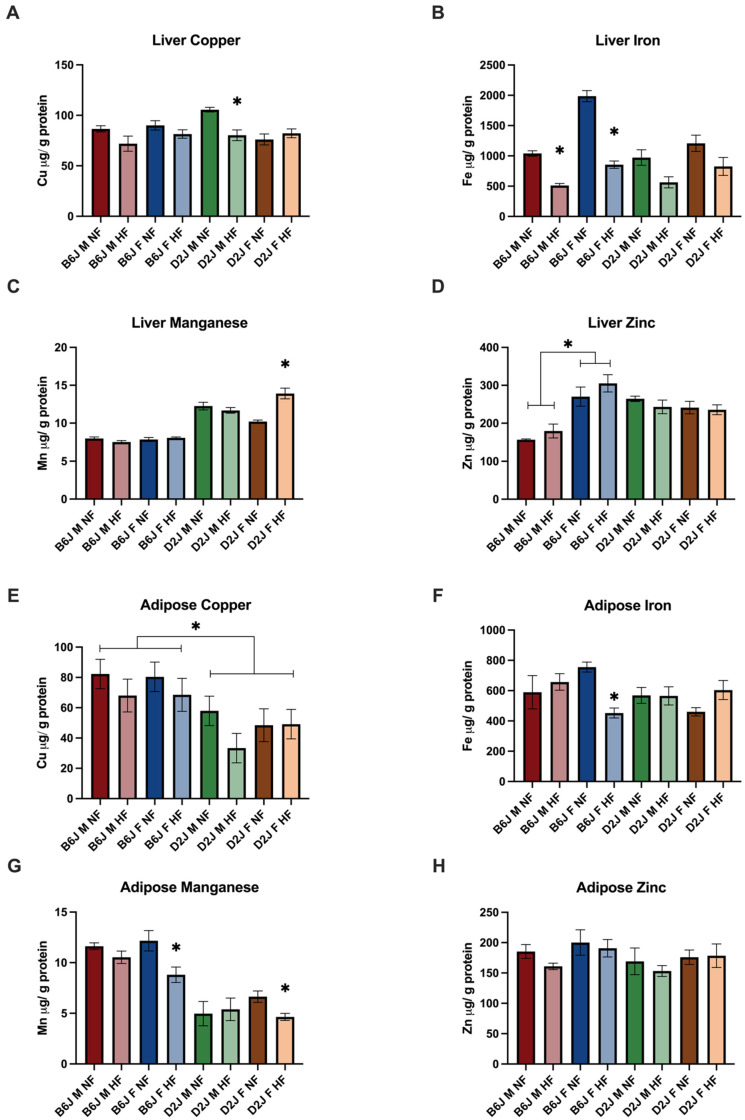
Liver and adipose trace element concentration by diet, sex, and strain. Atomic absorption spectrometry was used to determine copper, iron, manganese, and zinc concentration in µg of element/dL of tissue. (**A**) Liver copper concentrations were significantly decreased in D2J males (* *p* < 0.001). (**B**) There was a significant three-way interaction between sex, strain, and diet for liver iron concentrations, which were significantly reduced for B6J male and female mice (* *p* < 0.001). (**C**) Liver manganese concentrations were significantly increased in D2J females (* *p* < 0.001). (**D**) Liver zinc concentrations were significantly increased for B6J males and females (* *p* < 0.001 and *p* = 0.005). (**E**) There was a statistically significant main effect of strain on adipose copper concentration (* *p* = 0.001). (**F**) Adipose iron concentrations were significantly reduced in B6J males (* *p* = 0.002). (**G**) Adipose manganese concentrations were significantly decreased in B6J and D2J females (* *p* = 0.016 and *p* = 0.039). (**H**) There were no strain, sex, or sex interactions effects for zinc in adipose tissue. The asterisk (*) indicates statistical significance between normal-fat (NF) and high-fat (HF) diet groups. Data are represented as mean ± SEM. B6J M NF = B6J male normal fat, B6J M HF = B6J male high fat, B6J F NF = B6J female normal fat, B6J F HF = B6J female high fat, D2J M NF = D2J male normal fat, D2J M HF = D2J male high fat, D2J F NF = D2J female normal fat, D2J F HF = D2J female high fat.

**Figure 3 ijms-23-13778-f003:**
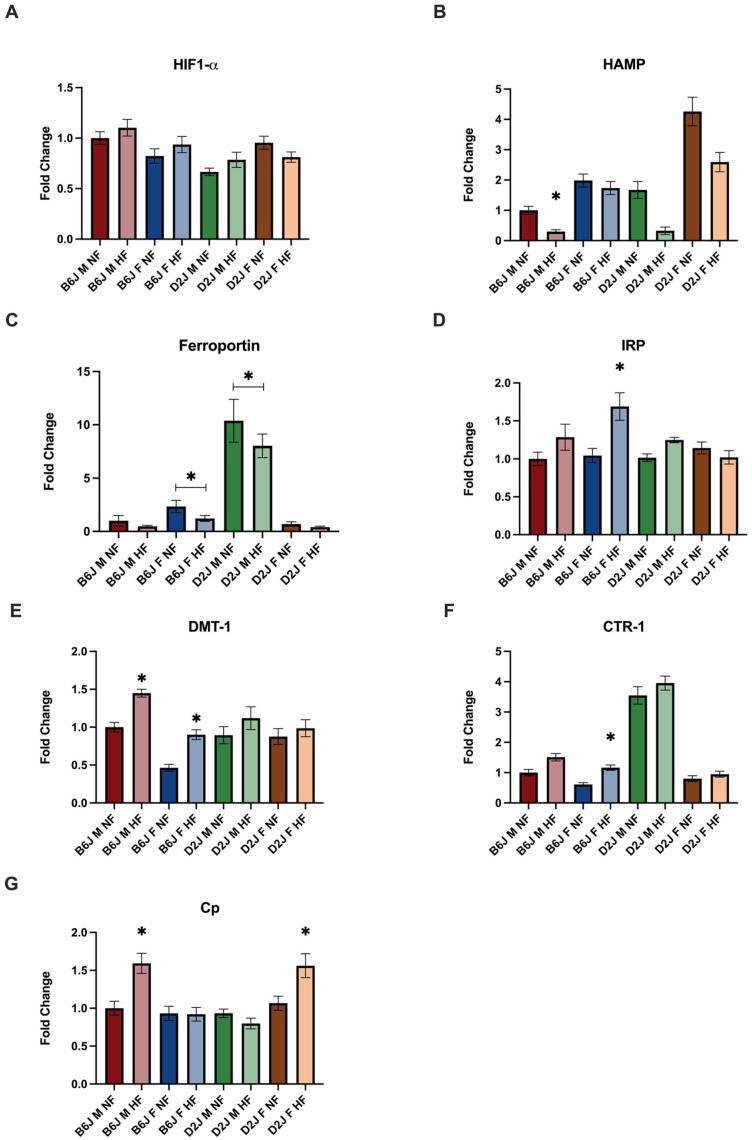
The effect of diet, sex, and strain on gene expression of proteins involved in trace element homeostasis in the liver. Gene expression is expressed as fold change normalized to the B6J male normal-fat groups are shown for (**A**) hypoxia inducible factor-α, (**B**) hepcidin, (**D**) iron regulatory protein-1, (**E**) divalent metal trasporter-1, (**F**) high affinity copper uptake protein-1, and (**G**) ceruloplasmin. Significant effect of high fat diet compared to normal fat group is denoted. Gene expression is expressed as fold change normalized to B6J males is shown for (**C**) ferroportin for which significant effect of sex and strain is denoted. Data are represented as mean ± SEM, * *p* < 0.05. B6J M NF = B6J male normal fat, B6J M HF = B6J male high fat, B6J F NF = B6J female normal fat, B6J F HF = B6J female high fat, D2J M NF = D2J male normal fat, D2J M HF = D2J male high fat, D2J F NF = D2J female normal fat, D2J F HF = D2J female high fat.

**Figure 4 ijms-23-13778-f004:**
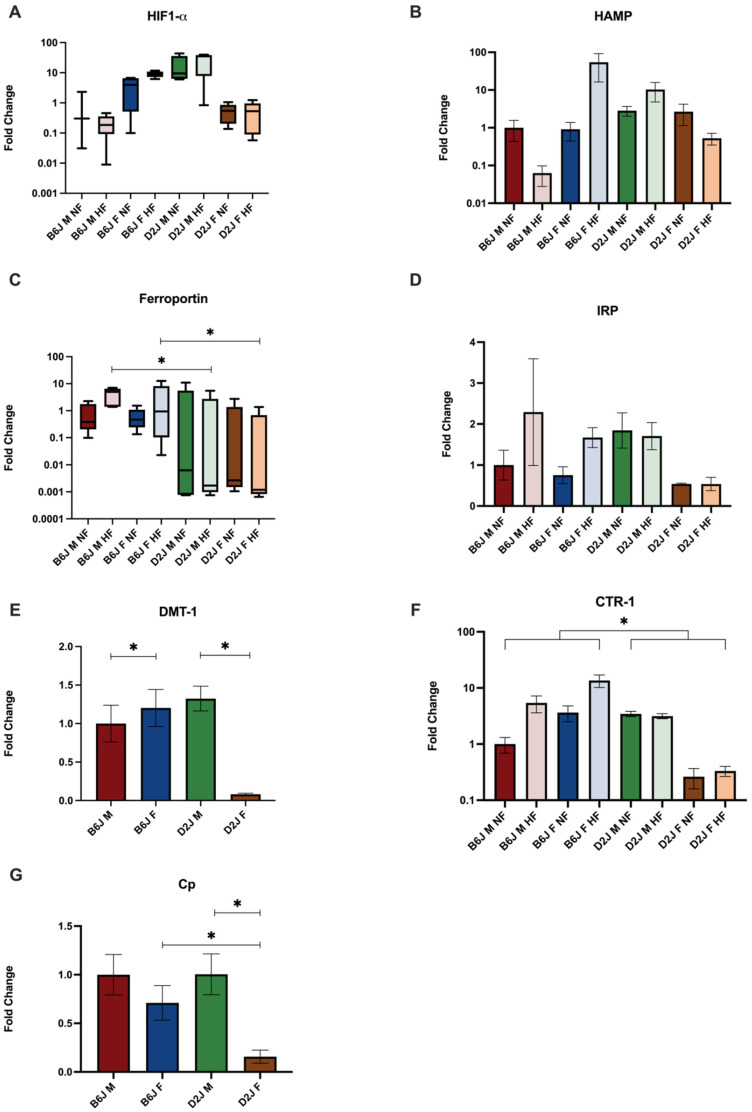
The effect of diet, sex, and strain on gene expression of proteins involved in trace element homeostasis in adipose tissue. Gene expression is expressed as fold change normalized to the B6J male normal-fat groups are shown for (**A**) hypoxia inducible factor-α, (**B**) hepcidin, (**C**) ferroportin, (**D**) iron regulatory protein-1, (**E**) divalent metal trasporter-1, (**F**) high affinity copper uptake protein-1, and (**G**) ceruloplasmin. Significant effect of high fat diet compared to normal fat group is denoted. Data are represented as mean ± SEM for B, D-G and median, 25th, 75th percentiles, minimum and maximum for A and C. * *p* < 0.05. B6J M NF = B6J male normal fat, B6J M HF = B6J male high fat, B6J F NF = B6J female normal fat, B6J F HF = B6J female high fat, D2J M NF = D2J male normal fat, D2J M HF = D2J male high fat, D2J F NF = D2J female normal fat, D2J F HF = D2J female high fat, B6J M = B6J male, B6J F= B6J female, D2J M = D2J male, D2J F = D2J female.

**Figure 5 ijms-23-13778-f005:**
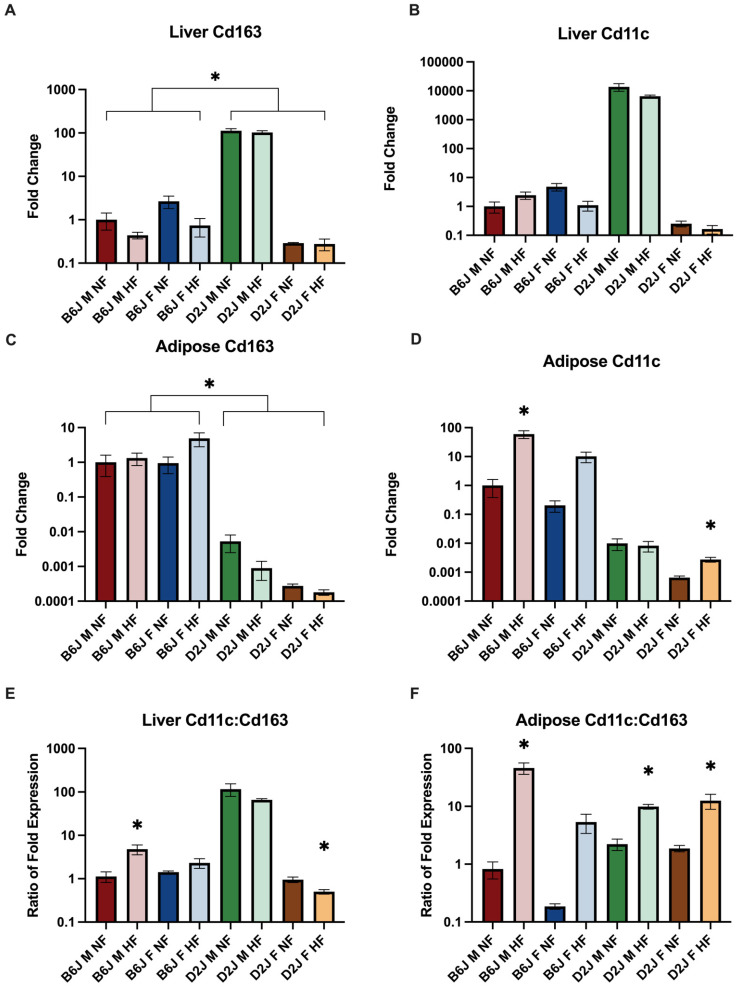
The effect of diet, sex, and strain on gene expression of repair and inflammatory macrophage phenotypes in liver and adipose tissue. Gene expression is expressed as fold change normalized to the B6J male normal fat biological group for (**A**) liver Cd163, (**B**) liver Cd11c, (**C**) adipose Cd163, and (**D**) adipose Cd11c. Ratio of Cd11c to Cd163 expression is shown for (**E**) liver and (**F**) adipose. Significant effect of high fat diet compared to normal fat group is denoted. Data are represented as mean ± SEM, * *p* < 0.05. B6J M NF = B6J male normal fat, B6J M HF = B6J male high fat, B6J F NF = B6J female normal fat, B6J F HF = B6J female high fat, D2J M NF = D2J male normal fat, D2J M HF = D2J male high fat, D2J F NF = D2J female normal fat, D2J F HF = D2J female high fat.

**Figure 6 ijms-23-13778-f006:**
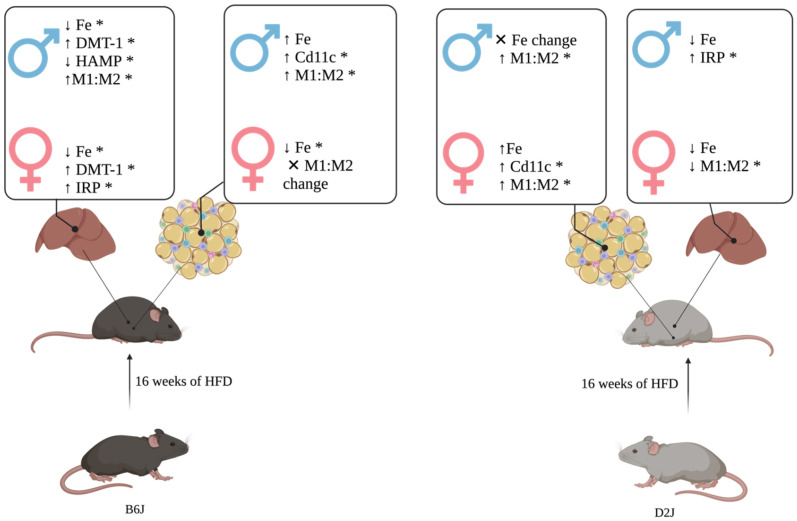
Summary model of the impact of a high-fat diet, sex, and strain on adipose and liver tissue iron concentration and iron-related gene expression. Following 16 weeks of a high-fat diet, B6J and D2J males and females experienced a significant weight gain and demonstrated significant alterations to adipose tissue and liver iron concentrations and related gene expression. Given the relevance of tissue iron concentrations to the development of local insulin resistance, our model focuses on HFD-induced alterations to tissue iron homeostasis. * *p* < 0.05. ↑ = increased, ↓  = decreased, Fe = iron, DMT-1 = divalent metal transporter 1, hepcidin= hepcidin, IRP = iron regulatory protein, M1:M2 = ratio of M1-like to M2-like macrophage gene expression. Figure created with BioRender.com.

**Table 1 ijms-23-13778-t001:** Feed Efficiency Rate: Feed efficiency rate in kcal consumed per gram of weight gain per week. Feed efficiency rate was averaged for B6J male, B6J female, D2J male, and D2J female high-fat and normal-fat groups over the 16-week 60% kcal from fat high-fat diet or 10% kcal from fat normal-fat diet intervention. There were no significant differences between the average feed efficiency rate amongst NFD (6.559 ± 5.316 kcal/gm weight gained) and HFD (8.021 ± 1.367 kcal/gm weight gained) groups (t(12.449) = 0.266, *p =* 0.794), B6J (10.021 ± 4.461 kcal/gm weight gained) and D2J (4.560 ± 2.996 kcal/gm weight gained) strains (t(22) = 1.016, *p =* 0.321), and NFD and HFD groups for B6J males (t(4) = −2.172, *p =* 0.096), B6J females (t(2.003) = −0.064, *p =* 0.952), D2J males (t(2.134) = −1.016, *p =* 0.205), and D2J females(t(4) = −0.619, *p =* 0.569).

Strain	Sex	Diet																	
			Week																
				2	3	4	5	6	7	8	9	10	11	12	13	14	15	16	Average
B6J	Male	Normal Fat		2.46 ± 0.12	7.53 ± 0.96	6.45 ± 1.52	15.66 ± 4.94	14.96 ± 1.73	13.15 ± 2.60	14.81 ± 0.47	−0.40 ± 15.58	21.13 ± 9.09	28.40 ± 11.85	38.14 ± 64.80	10.99 ± 1.96	14.18 ± 1.36	57.41 ± 53.23	23.99 ± 20.88	17.93 ± 4.59
B6J	Male	High Fat		2.05 ± 0.17	5.58 ± 0.72	5.78 ± 0.47	4.99 ± 0.35	5.37 ± 0.44	4.67 ± 0.29	5.40 ± 0.57	5.46 ± 0.54	5.28 ± 0.54	6.90 ± 0.77	8.56 ± 2.05	10.05 ± 2.80	12.22 ± 2.40	19.29 ± 2.12	17.08 ± 11.49	7.91 ± 0.46
B6J	Female	Normal Fat		4.21 ± 0.26	6.36 ± 0.91	9.63 ± 1.33	22.44 ± 7.96	20.59 ± 5.51	93.79 ± 102.42	82.23 ± 83.65	−71.24 ± 108.48	16.06 ± 3.00	−2.42 ± 33.33	−15.98 ± 70.66	12.11 ± 21.62	−2.38 ± 29.74	−2.74 ± 16.09	−75.12 ± 111.54	6.50 ± 19.337
B6J	Female	High Fat		3.00 ± 0.05	5.32 ± 0.80	10.75 ± 2.15	9.61 ± 1.68	7.66 ± 1.70	6.62 ± 1.89	6.43 ± 0.67	6.02 ± 0.16	7.89 ± 1.71	7.99 ± 1.43	10.24 ± 2.92	8.24 ± 0.70	8.42 ± 1.13	8.63 ± 1.65	9.32 ± 0.81	7.74 ± 0.54
D2J	Male	Normal Fat		2.71 ± 0.14	6.54 ± 3.19	6.18 ± 1.12	9.39 ± 3.03	−5.49 ± 23.71	11.33 ± 2.99	14.00 ± 0.98	15.13 ± 3.92	16.26 ± 6.95	−57.69 ± 151.16	−121.64 ± 110.97	28.06 ± 14.74	45.04 ± 69.20	2.18 ± 30.39	56.68 ± 24.26	1.91 ± 10.29
D2J	Male	High Fat		1.98 ± 0.08	3.11 ± 0.18	3.76 ± 0.38	3.96 ± 0.40	6.05 ± 0.58	6.47 ± 0.72	7.37 ± 1.28	12.17 ± 3.49	7.32 ± 6.66	7.92 ± 3.12	36.27 ± 22.68	11.03 ± 2.32	11.93 ± 2.26	34.51 ± 17.07	34.21 ± 24.61	12.54 ± 1.88
D2J	Female	Normal Fat		3.31 ± 0.35	7.57 ± 2.36	17.64 ± 15.66	9.51 ± 2.27	10.25 ± 3.28	12.37 ± 31.05	9.74 ± 3.26	−26.82 ± 12.04	25.01 ± 18.88	−35.43 ± 174.47	−1.25 ± 31.56	−27.71 ± 24.56	23.40 ± 9.28	15.94 ± 20.34	−45.10 ± 46.62	−0.11 ± 4.84
D2J	Female	High Fat		2.62 ± 0.08	3.05 ± 0.32	4.19 ± 0.25	4.85 ± 0.35	11.66 ± 6.09	9.17 ± 3.26	6.33 ± 1.24	10.71 ± 4.50	8.93 ± 1.03	−11.26 ± 34.95	25.11 ± 30.06	31.97 ± 23.42	3.90 ± 22.97	−47.94 ± 26.12	−4.89 ± 42.44	3.89 ± 4.28

## Data Availability

Data associated with this manuscript may be requested from the senior author of this paper. Professor Keith Erikson; kmerikso@uncg.edu.

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
