# Peer review of "Influence of Sex and Strain on Hepatic and Adipose Tissue Trace Element Concentrations and Gene Expression in C57BL/6J and DBA/2J High Fat Diet Models"

_ijms, 2022, doi:10.3390/ijms232213778_

Round 1

Reviewer 1 Report

Comments to the Authors 

This review titled “Influence of sex and strain on hepatic and adipose tissue trace element concentrations and gene expression in C57BL/6J and DBA/2J high fat diet models”, mainly reported the influence of sex and strain on the dysregulation of trace element concentration and associative gene expression due to diet induced obesity in adipose tissue and the liver.  In general, the manuscript is well organized and language is also OK. The authors has previously published the related studies about HFD leading to the dysregulation of trace element concentration and gene expression in several regions of the brain. Therefore, this article can be accepted after minor revision.

 Some small points are shown below.

1. The meanings or values of this research should be added in Section Introduction.

2. The authors should revise the abscissa in Fig 2. Fig.3. with same abbreviation.

B6J M NF= B6J male normal fat, B6J M HF= B6J male high fat, B6J F NF= B6J female normal fat, B6J F HF= B6J female high fat, D2J M NF= D2J male normal fat, D2J M HF= D2J male high fat, D2J F NF= D2J female normal fat, D2J F HF= D2J female high fat.

3.  The quality of all figures need to be improved.

Author Response

Response to Reviewer 1 Comments

The authors thank Reviewer 1 for their time, careful consideration of our manuscript, and comments.

Point 1: The meanings or values of this research should be added in Section Introduction.

Point 1 Response: Thank you for this comment. In the introduction, we have amended the statement regarding the gap in the literature that our research addresses. In this statement we identify the value of the research. “However, the influence of sex and strain on the systemic distribution of trace elements and gene expression in a model of DIO, remains unexplored and valuable in the context of metabolic conditions associated with altered trace element homeostasis, such as tissue insulin resistance.” This is discussed in further detail in the discussion and conclusion sections of the manuscript.

Point 2: The authors should revise the abscissa in Fig 2. Fig.3. with same abbreviation.

B6J M NF= B6J male normal fat, B6J M HF= B6J male high fat, B6J F NF= B6J female normal fat, B6J F HF= B6J female high fat, D2J M NF= D2J male normal fat, D2J M HF= D2J male high fat, D2J F NF= D2J female normal fat, D2J F HF= D2J female high fat.

Point 2 Response: Figure 2 graph E was depicted as Copper µg/gm protein for each strain (B6J and D2J) in order to visually represent the significant strain differences. Similarly, in Figure 3 graph C was depicted as fold change for each sex/strain group (B6J male, B6J female, D2J male, D2J female) given the significant differences for these groups rather than differences by diet, as depicted in the other graphs within the figure. For consistency, the abscissa for Figures 2E and 3C have been revised to be consistent with other graphs in the figure with the same abbreviations (B6J M NF= B6J male normal fat, B6J M HF= B6J male high fat, B6J F NF= B6J female normal fat, B6J F HF= B6J female high fat, D2J M NF= D2J male normal fat, D2J M HF= D2J male high fat, D2J F NF= D2J female normal fat, D2J F HF= D2J female high fat).

Point 3: The quality of all figures need to be improved.

Point 3 Response: The figures have been changed from JPEG File Interchange Format (JPG) to Portable Network Graphics (PNG) format and the resolution of all figures have been increased to 1200 dpi.

Reviewer 2 Report

Was the weight of the mice under investigation measured during the start and at tissue collection? If yes, the authors should include a line regarding this information.

Author Response

Response to Reviewer 2 Comments

The authors thank Reviewer 2 for their time, careful consideration of our manuscript, and comments.

Point 1: Was the weight of the mice under investigation measured during the start and at tissue collection? If yes, the authors should include a line regarding this information.

Point 1 Response: Yes, the weight of all mice was taken initially, at the start of the prescribed diet, then weekly during diet intervention, and finally, during week 16 prior to tissue collection. We have added a sentence to clarify this point in subsection 2.1 Animals and Diet of the Methods section.
